# Rational Use of Danofloxacin for Treatment of *Mycoplasma gallisepticum* in Chickens Based on the Clinical Breakpoint and Lung Microbiota Shift

**DOI:** 10.3390/antibiotics11030403

**Published:** 2022-03-17

**Authors:** Shuge Wang, Anxiong Huang, Yufeng Gu, Jun Li, Lingli Huang, Xu Wang, Yanfei Tao, Zhenli Liu, Congming Wu, Zonghui Yuan, Haihong Hao

**Affiliations:** 1National Reference Laboratory of Veterinary Drug Residues (HZAU) and MAO Key Laboratory for Detection of Veterinary Drug Residues, Wuhan 430070, China; shugewang@webmail.hzau.edu.cn (S.W.); anxionghuang@webmail.hzau.edu.cn (A.H.); guyufeng@webmail.hzau.edu.cn (Y.G.); huanglingli@mail.hzau.edu.cn (L.H.); wangxu@mail.hzau.edu.cn (X.W.); tyf@mail.hzau.edu.cn (Y.T.); liuzhli009@mail.hzau.edu.cn (Z.L.); yuan5802@mail.hzau.edu.cn (Z.Y.); 2MOA Laboratory for Risk Assessment of Quality and Safety of Livestock and Poultry Products, Wuhan 430070, China; 3National Center for Veterinary Drug Safety Evaluation, College of Veterinary Medicine, China Agricultural University, Beijing 100193, China; wucm@cau.edu.cn; 4Institute of Food Safety and Nutrition, Jiangsu Academy of Agricultural Sciences, Nanjing 210014, China; lijunjaas@126.com

**Keywords:** danofloxacin, *Mycoplasma gallisepticum*, epidemiological cutoff values, PK–PD cutoff values, clinical cutoff values, clinical breakpoint, lung microbiota

## Abstract

The study was to explore the rational use of danofloxacin against *Mycoplasma gallisepticum* (*MG*) based on its clinical breakpoint (CBP) and the effect on lung microbiota. The CBP was established according to epidemiological cutoff value (ECV/CO_WT_), pharmacokinetic–pharmacodynamic (PK–PD) cutoff value (CO_PD_) and clinical cutoff value (CO_CL_). The ECV was determined by the micro-broth dilution method and analyzed by ECOFFinder software. The CO_PD_ was determined according to PK–PD modeling of danofloxacin in infected lung tissue with Monte Carlo analysis. The CO_CL_ was performed based on the relationship between the minimum inhibitory concentration (MIC) and the possibility of cure (POC) from clinical trials. The CBP in infected lung tissue was 1 μg/mL according to CLSI M37-A3 decision tree. The 16S ribosomal RNA (rRNA) sequencing results showed that the lung microbiota, especially the phyla *Fi**rmicutes* and *Proteobacteria* had changed significantly along with the process of cure regimen (the 24 h dosing interval of 16.60 mg/kg b.w for three consecutive days). Our study suggested that the rational use of danofloxacin for the treatment of *MG* infections should consider the MIC and effect of antibiotics on the respiratory microbiota.

## 1. Introduction

*Mycoplasma gallisepticum* (*MG*), mainly causes sinusitis, bronchitis, and air sacculitis, and leads to chronic respiratory diseases (CRD), with reduced chicken immunity and increases the chance of co-infection with other pathogens [1,2]. *MG* produces a variety of metabolites that cause dysfunction of the respiratory mucosal epithelial cells. It can also migrate to the lungs and air sacs, leading to lung and balloon lesions, resulting in effusion [3,4]. Typical symptoms of *MG* infection are severe weight loss, coughing, runny nose, and a vocal sound when breathing, with a slow and long course [5]. The disease can decrease the growth rate of the chicken and the conversion rate of the feed to bring huge economic loss to poultry farming [6,7,8].

Danofloxacin is a third-generation animal-specific fluoroquinolone developed by Pfizer Inc. of the United States [9]. Compared with other antibacterial drugs, danofloxacin have stronger cell permeability, higher drug concentration in plasma and tissues, and stronger antibacterial activity that can exhibit an antibacterial effect even when the drug concentration is low [10]. Danofloxacin is widely used in the treatment of respiratory diseases caused by *Mycoplasma*, *A**ctinobacillus* and *G**laesserella parasuis* [11,12,13]. Therefore, it is expected to have potentially wide applications for controlling respiratory tract disease caused by *MG* [14,15,16].

Clinical studies have shown that chicken CRD caused by *MG* can be cured by danofloxacin, and few resistant strains emerged [17]. Pharmacokinetic–pharmacodynamic (PK–PD) models reflect the relationship among the drug, bacteria, and the infected animal, and the model also quantifies the effects of antimicrobials on target pathogens at the same time [18,19]. In general, the concentration of drugs in plasma is relatively easy to measure compared to tissue or lavage fluid [20]. However, it has been reported that for non-mixed infections, the concentration of free drug in the target tissue is the determining factor for treating local infections [21]. The PK data in the target tissue is more convincing than the data in the plasma; a reasonable dosing regimen is given by this data using WinNonlin and Monte Carlo analysis [22]. Previous studies have reported the methods for establishing a PK–PD model by drug concentrations in target tissues to replace drug concentrations in plasma [23].

The breakpoints play an important role in clinical microbiology and are used to define the sensitivity and resistance of strains to antimicrobial agents [24]. The main factors considered in the formulation of the breakpoint include the epidemiological cut-off value (ECV), the PK and PD data in the target animal, the MIC of clinical pathogenic bacteria in clinical effectiveness trials, and prognostic results on the clinical and bacteriological level [25]. These are defined in terms of the ECV, the clinical cutoff value (CO_CL_), and the PK–PD cutoff value (CO_PD_) [26,27]. The ECV is the upper boundary of the wild-type MIC distribution for organisms that do not have detectable acquired resistance mechanisms [28]. The PK–PD cutoff is defined as the highest possible MIC for which a given percentage of target animals (usually 90%) achieve a defined PK–PD parameter value (e.g., PK–PD parameter value for a bactericidal effect) [29]. The CO_CL_ is determined according to the relationship between clinical outcomes and antimicrobial susceptibility by statistical approaches [27].

The 16S rRNA gene sequencing technique has become the most widely used method to investigate the composition of microbial ecosystems of digestive and respiratory tracts in recent years [30,31,32,33]. There are some evidence that the consists of the respiratory microbiota of chickens depend on the feed, living environment, and immunity of chickens [34,35,36]. However, no studies have shown the influences on the lung microbiota after administration of the drug. Consequently, it is very necessary to get a picture of the effect of antibiotics used on the lung microbiota for the selection of the most effective treatment method.

This study established CBP based on CLSI and EUCAST, in order to standardize rational and precise use of danofloxacin for the treatment of *MG* and minimize the emergence of danofloxacin resistance. In addition, the effect of danofloxacin in the treatment of *MG* infection on chicken lung microbiota was analyzed by 16S rDNA sequencing.

## 2. Materials and Methods

### 2.1. Chemicals and Reagents

A standard Danofloxacin drug sensitivity test (>94.2%) was purchased from Dr. Ehrenstorfer (Augsburg, Germany). Danofloxacin mesylate raw material powder for animal experiments (≥98.5%) was purchased from Dalian Ronghai Biotechnology Co., Ltd. (Dalian, Ronghai, China). All the chemical reagents and organic solvents were analytical reagents (AR) or HPLC grade. Ampicillin (≥98%) was purchased from Beijing Solarbio Technology Co., Ltd. (Beijing, China). Hydrolyzed milk protein was purchased from BIOSHARP Biotechnology Co., Ltd. (Beijing, China). DMEM high glucose was obtained from Shanghai Hengfei Biological Technology Co., Ltd. (Shanghai, China). The method of the medium for isolation and culture of *MG* is described in a patent application (number 201910323646.6).

The preparation of FM-4 base fluid was as follows: 2.5 g NaCI, 0.2 g KCI, 0.8 g Na_2_HPO_4_·12H_2_O, 0.1 g MgSO_4_·7H_2_O, 0.05 g KH_2_PO_4_, 5.0 g glucose, 2.5 g hydrolyzed milk protein, 2.5 g yeast extract powder and 400 mL deionized water were mixed, autoclaved at 105 °C for 20 min and finally stored at 4 °C for later use.

To prepare FM-4 broth, 80 mL of healthy inactivated fetal bovine serum, 80 mL of DMEM high glucose, 0.5 mL of penicillin (1 million IU/mL), and 1 mL of phenol red (1%) were added to 400 mL of modified FM-4 base solution. In order to ensure that *MG* grows more quickly, 4 mL of 1% Arginine, 4 mL of 1% Cysteine, and 4 mL of 1% Coenzyme I (NAD) were added.

The preparation of FM-4 agar plates was as follows. Before sterilization, 1.0~1.2% agar was added to the modified FM-4 base solution, adjusting the pH to 8.0 (±0.2) with 1 mol/L NaOH, boiling to fully dissolve, and autoclaving at 105 °C for 20 min.

### 2.2. Isolation and Cultivation of MG

The 111 *MG* strains were collected from clinical samples and deposited in the National Reference Laboratory of Veterinary Drug Residues in Huazhong Agricultural University. The chicken respiratory tract cotton swabs were processed aseptically placed in the FM-4 medium for enrichment culture in a 37 °C, 5% CO_2_ incubator for 3–5 days until the culture changed from red to yellow. The culture was then filtered (0.22 μm) and plated on an FM-4 agar plate. All bacterial isolates were confirmed by polymerase chain reaction (PCR). The primer sequences for the identification of *MG* are shown in Table 1.

The reaction conditions were pre-denaturation at 94 °C for 5 min, denaturation at 94 °C for 45 s, annealing at 55 °C for 45 s, extension at 72 °C for 1 min, 34 cycles of amplification, and final extension at 72 °C for 5 min, and the amplified products were analyzed by 1.5% agarose gel electrophoresis.

### 2.3. Animals

A total of 130 three-week-old broilers (half male and female) with an average weight of 500~550 g were offered for the PK study and the other 165 broilers for the clinical trials. These broilers were placed in different cages with good ventilation and warmth, free access to drinking water, and fed to a group diet that did not contain any antibiotics and additives. The use of chickens and all experimental protocols in this study were approved by the Institutional Animal Care and Use Committee at HZAU (approval number HZAUSW-2018-046). All procedures regarding animal care and testing were carried out according to the recommendation for the care and use of laboratory animals of Hubei provincial public service facilities.

### 2.4. Determination of ECV

#### 2.4.1. Antimicrobial Susceptibility Determination

The sensitivity was determined by the color change unit (CCU) of the medium [37]. The CCU of the strain was determined by tenfold dilutions using the FM-4 medium, and the highest dilution that resulted in color change was the CCU of the strain [38]. *E. coli* (ATCC 25922) was selected as the QC strain.

The MIC of danofloxacin to *MG* was determined by the micro broth dilution method. The bacterial solution was diluted with the modified FM-4 liquid medium until the titer was 10^5^ CCU/mL, and then the bacteria diluent was added to the bacterial culture plate. Positive control and negative control were set up in the 96-well plate at the same time. After adding the sample, the plate was sealed with sterile medical tape and placed in a 37 °C incubator for five days.

MIC judgment method was when the control group with added bacteria were yellow, and the color of the negative control wells did not change, recording the MIC of the well without color change, which was the MIC of danofloxacin for the strain. The MIC measurement was repeated three times.

#### 2.4.2. Formulation of ECV

The MIC, MIC_50_, and MIC_90_ of danofloxacin against *MG* were analyzed. The obtained MIC data were simulated by the nonlinear regression method proposed by Turnidge et al. [28] and the ECOFFinder software [39,40].

### 2.5. Establishment of CO_PD_

#### 2.5.1. Selection of Pathogenic Strain M19

The pathogenic strains of *MG* with MIC value at MIC_90_ and the healthy SPF (Specific Pathogen Free) embryos of 5–7 days were selected for this study. *MG* was injected into the yolk sac of the chicken embryo after the strains had grown to the logarithmic phase (1 × 10^9^ CCU/mL). Each strain was one group, each group had 10 chicken embryos, and the control group (FM-4 medium) was set up for the control. The volume of *MG* was 0.5 mL for each chicken embryo. After injection, the chicken embryos were cultured in an incubator with a constant temperature of 37 °C, and an amount of moisture was needed. The survival of infected chicken embryos was also observed and counted, and the pathological symptoms of dead and the blank control chicken embryo were also observed for one week. The *MG* strains were isolated and identified from the infected chicken embryos to ensure the success of the infection test.

#### 2.5.2. PD Study of M19 In Vitro and Ex Vivo

The MIC and MBC for the M19 strain were determined in vitro and ex vivo (plasma and lung tissue) by the micro-dilution method as mentioned above. The mutant prevention concentration (MPC) of danofloxacin was determined using the agar dilution method. For each of the *MG* strains, the volume of 100 μL of culture with the CCU of 10^10^ CCU/mL was inoculated onto the FM-4 agar plates containing serial dilutions of danofloxacin (1 to 32 MIC). MPC was defined as the lowest concentration that inhibited bacterial growth for 168 h at 37 °C. The experiment was repeated three times.

The in vitro and ex vivo killing curves of danofloxacin in FM-4 and lung tissue were drawn by monitoring the CCU changes during the incubation of M19 under a series of danofloxacin concentrations (1/2 to 32 MIC) for a continuous time period (0, 4, 8, 12, 16, 20, 24, 28, 32, 36, 40, 44 and 48 h). The killing curves were determined in triplicate for each concentration.

#### 2.5.3. PK Studies of Danofloxacin in Chicken Plasma and Lung

The isolated M19 strain was selected for PK studies according to the chicken embryo infection test results (Appendix A). The 130 broilers were randomly and equally divided into Group A and Group B. Group A and Group B were divided into 13 groups, respectively, 5 in each group. Group A was given 5 mg/kg b.w at a single dose by gavage without treatment. Group B was challenged with M19 for about one week until the flock had significant respiratory symptoms and then given 5 mg/kg b.w at a single dose by gavage at 5 mg/kg b.w. Five broilers were killed in each group at every sampling time point. Then blood samples (5 mL) and lung samples from broilers were obtained at 0, 0.25, 1, 1.5, 2, 4, 6, 8, 12, 24, 36, and 48 h after administration.

#### 2.5.4. High-Performance Liquid Chromatography (HPLC) Method for Danofloxacin Determination

The danofloxacin present in the plasma and lung tissue samples after extraction were analyzed using the Waters 2695 series HPLC and a Waters 2487 UV detector set at a wavelength of 283 nm. An Agilent Agilent SB-Aq column (250 × 4.6 mm i.d., 5 μm; Agilent Technologies, Santa Clara, CA, USA) was used for separation. The mobile phases were 0.05% phosphoric acid and acetonitrile. The injection volume and flow rates were 20 μL and 0.8 mL/min, respectively. The peak time of danofloxacin was 10.64 min, and there were no endogenous interferences on the chromatograms.

#### 2.5.5. The Protein Binding Rate

The protein binding rate substitutes the result of the liquid phase test into the formula, using Equation (1):Fb = (Dt − Df)/Dt × 100%(1)
where Dt represents the drug concentration (total concentration) of the lung tissue in the dialysis bag; Df represents the drug concentration in the buffer (free drug concentration) outside of the dialysis bag.

#### 2.5.6. PK Analysis

PK parameters were calculated from the concentration of danofloxacin in plasma and lung tissue by WinNonlin software (version 5.2.1, Pharsight Corporation, Mountain View, CA, USA). A two-compartment model in WinNonlin software obtained the PK parameters according to the characteristics of the concentration-time figures.

#### 2.5.7. PK–PD Integration and Modeling

The determination of the time-killing curve ex vivo was as follows. The determination method of bactericidal curve ex vivo was the same as in vitro, except that the medium was replaced with plasma or lung fluid. The ex vivo killing-time curve was fitted to a PD model by the hypothesis that a decrease in danofloxacin concentration is based on incubation time with the inhibitory sigmoid E_max_ model.

According to the ex vivo time-killing curve, the Sigmoid Emax model ($E=E_{\max}-\frac{\left(E_{\max}-E_{0}\right)C^{N}}{C^{N}+{\mathrm{EC}}_{50}^{N}}$) was used to calculate the AUC_24_/MIC (AUIC) of danofloxacin at different concentrations, C represented AUC_24_/MIC, which indicated the bacteriological outcome of AUC over 24 h were calculated using MIC in vitro and PK parameters in vivo. E was a difference of antibacterial CCU logarithm of lung samples incubated with drug; E_max_ was a maximum difference of antibacterial CCU logarithm of lung samples incubated with drug; E_0_ was the difference after 24 h incubation in CCU antibacterial logarithm in control samples; EC_50_ was the PK–PD parameter value ex vivo when the 50% maximal bactericidal effect was produced in the lung tissue sample; N was the Hill coefficient, which was used to describe the slope of the PK–PD parameter value and the effect E linearization ex vivo, and determined the S-shaped curve.

#### 2.5.8. The Monte Carlo Analysis and Establishment of CO_PD_

A total of 10,000 iterations by Monte Carlo simulation (MCS) were conducted using Oracle Crystal Ball software (version 7.2.2) (Oracle Corporation, Redwood Shores, CA, USA) and was based on PK parameters and calculated PK–PD targets of lung tissues (AUC_24h_/MIC) for bactericidal action (E = −3) [41]. The AUC_24h_ was presumed to be log-normally distributed for the confidence intervals (CI) and mean values. The PK–PD target was evaluated. Conservative value (AUC/MIC = 37.34) was selected to calculate the probability of target attainment (PTA). CO_PD_ was defined as the MIC when the PTA was ≥90%.

### 2.6. Exploration of CO_CL_

#### 2.6.1. Dose Regimen Based on PK–PD Model

Calculation of the dosage under different parameters used the dosage Equation (2),
(2)$$\mathrm{Dose}=\frac{\mathrm{CL}\times \mathrm{MIC}\times {\left({\mathrm{AUC}}_{24}/\mathrm{MIC}\right)}_{\mathrm{ex}}}{F\times fu}$$
where CL refers to the clearance rate of danofloxacin in the chicken respiratory tract; (AUC_24_/MIC)_ex_ refers to the PK–PD parameter value corresponding to different therapeutic effects (bactericidal, antibacterial and eradication); MIC is the minimum inhibitory concentration of M19 strain; F is the bioavailability; *fu* is the free drug concentration ratio.

The probabilities of distribution for daily doses were run for 10,000 trials to achieve the target value of 90% for (E = 0) bacteriostatic, (E = −3) bactericidal, and (E = −4) bacterial elimination effects.

#### 2.6.2. Infection Model and Clinical Trials

A total of 165 healthy broilers (500 ± 50 g) were divided into 11 groups: five groups were the experimental group, five groups were the negative control group, and one group was the blank control group, with 15 broilers in each group. The five experimental groups and five negative control groups were challenged with five representative strains, M2, M19, M24, M57, and M73, by tracheal perfusion inoculation of 1 × 10^10^ CCU bacterial suspension once a day for 7 d. The blank control group was inoculated with blank FM-4 broth. The dosage regimens were recommended by the PK–PD therapeutic dosage regimen. After oral administration, these broilers were monitored daily for one week. The cure rate and mortality were counted. After successful infection, the clinical bacteriology prognosis was determined by the real-time fluorescence quantitative PCR method by extracting the throat swab DNA of each group of chickens at regular intervals every day during the all experiment period.

#### 2.6.3. The Establishment of the Standard Curve of *MG* by Real-Time PCR

Strain and samples. *MG* S6 was obtained from the National Reference Laboratory of Veterinary Drug Residues. *Salmonella* ATCC 14028, *Klebsiella pneumoniae* CMCC 46117, *Pseudomonas aeruginosa* ATCC 27853, *Staphylococcus aureus* ATCC 6538 were purchased from Huankai Microbial Technology Co., Ltd. (Guangdong, China). *Pasteurella pneumotropica* ATCC 35149, *Corynebacterium kutscheri* ATCC 15,677 were purchased from American Type Culture Collection (Manassas, VA, USA).

Throat swabs from clinical trials were sampled as described [42], and transferred to the laboratory for real-time PCR testing. One throat swab per bird was used for real-time PCR as follows: First, the tracheal swab was streaked onto FM-4 agar and then placed into 400 µL of sterile physiologic saline. After vortexing for 2 min, the mixture was used for DNA isolation.

DNA extraction. DNAs from *MG* S6 and clinical samples were extracted with a commercial DNA isolation kit (QIAMP DNA mini kit, Berlin, Germany).

Primers. Forward and reverse PCR primers, *MG*-1F (5′-GATTTCGAAGAATCAACTGT-3′) and *MG*-2R (5′-AAGGGATTAATATTCCCAAC-3′) were selected to amplify an internal portion of an *MG* lipoprotein gene. The expected amplicon size was 400 bp with Vector NTI 4.0 software (Infomax, Inc., North Bethesda, MD, USA).

Construction of plasmid standard. A standard plasmid pUC57-*MG* containing the sequence of the target fragment was synthesized by Beijing Kinco Xinye Biotechnology Co., Ltd. (Beijing, China). The obtained plasmid standard was amplified with the universal sequencing primers M13-F (5′-TGTAAAACGACGGCCAGT-3′) and M13-R (5′-CAGGAAACAGCTATGACC-3′) of vector pUC57 and sequenced by Beijing Kinco Xinye Biotechnology Co., Ltd.

Real-time PCR assay system and conditions. Real-time PCR was performed using a LightCyclerTM 2.0 (Roche Diagnostics, Mannheim, Germany). Each reaction had a volume of 20 µL, including 18 µL of reaction mixture containing 1 × LC FastStart DNA SYBR Green I Master Mix (Roche), MgCl_2_ (4 mM), a 0.5 µM concentration of each primer and 2 µL of template DNA.

Cycling parameters were as follows. Initial denaturation at 95 °C for 10 min followed by 40 cycles of denaturation at 95 °C for 10 s, annealing at 50 °C for 5 s and extension at 72 °C for 20 s. Melting curve analysis was performed automatically by LightCycler 2.0 software (Version 3), and the melting peaks were expected to occur at a melting temperature (Tm) of 80 °C.

Standard curve construction. Measurement of the concentration of the plasmid standard was performed with an ultra-micro UV spectrophotometer (Roche Molecular Biochemicals, Indianapolis, IN, USA).

Calculation of the copy concentration of the plasmid standard according to the Equation (3):(3)$$A\left(\mathrm{copies}/\mu L\right)=\frac{6.02\times 10^{23}\times C\left(\mathrm{ng}/\mu L\right)\times 10^{-9}}{L\left(\mathrm{bp}\right)\times 660}$$
where A represents the copies of plasmid standard. C represents the concentration of the sample. L represents the template length. Plasmid standard was diluted to 5 × 10^6^, 5 × 10^5^, 5 × 10^4^, 5 × 10^3^, 5 × 10^2^, 5 × 10^1^, 5 × 10^0^ copies/μL at a 10-fold ratio, and sterile PCR-grade deionized water is negative for comparison. A standard curve was drawn based on real-time fluorescence quantitative PCR amplification.

Sensitivity and specificity of real-time PCR. To verify the sensitivity of the method, six plasmid standards with a gradient of concentration dilutions (5 × 10^5^, 5 × 10^4^, 5 × 10^3^, 5 × 10^2^, 5 × 10^1^ and 5 × 10^0^ copies/μL) that were used as positive templates, sterile PCR-grade deionized water was used as a negative control for quantitative fluorescence PCR, respectively.

For specificity, real-time PCR was applied to the templates prepared from the following strains: *Salmonella* ATCC 14028, *Klebsiella pneumoniae* CMCC 46117, *Pseudomonas aeruginosa* ATCC 27853, *Staphylococcus aureus* ATCC 6538, *Pasteurella pneumotropica* ATCC 35149, *Corynebacterium kutscheri* ATCC 15677. These strains were selected for testing the specificity of real-time PCR.

Real-time PCR with clinical samples. The throat swabs were taken from each group of chickens at regular intervals every day during the experiment period. The same procedure was applied for DNA preparation from all of the samples. *MG* S6 strain DNA and sterile PCR-grade deionized water were used as positive and negative controls in each PCR run, respectively. Performing real-time PCR amplification and taking the number of cycles (Ct) of the reaction as the abscissa of the amplification curve of the PCR assay, and the intensity of the fluorescence signal could be seen as the ordinate to obtain the amplification curve and the standard curve.

#### 2.6.4. Statistical Analysis for the Establishment of CO_CL_

The obtained data were analyzed by WindoW, CART and nonlinear regression analysis to obtain the final CO_CL_. The WindoW approach, which constrains the determination of CO_CL_ to values contained within a window (or limited) range of MICs, included “MaxDiff” and “CAR”. The former represents the difference between higher and lower POC at a certain MIC. The latter was determined according to the clinical outcome and the corresponding MIC distribution. However, “CAR” could not be set as the lowest MIC or the highest MIC if “CAR” was gradually increasing with MIC, then the “CAR” should choose the second small “CAR”. The boundaries of WindoW were determined by two inflection points [27].

The second approach to establish CO_CL_ was the CART model (Salford Predictive Modeler software). MIC and POC were predictive variables and target variables, respectively. Moreover, the Gini coefficient minimization criterion was used to select the MIC node automatically.

Nonlinear regression analysis proposed by EUCAST was a new approach based on the formula between POC and MIC. Log_2_MIC was the independent variable, and the POC was the dependent variable. Frequently, the CO_CL_ was selected by the highest correlation coefficient of the model.

#### 2.6.5. Evaluation of the Effect of Danofloxacin on the Lung Microbiota of *MG* Infected Chickens

16S rRNA gene sequencing was used to analyze the effect of danofloxacin on the lung microbiota of *MG*-infected chickens. The effective tags of all samples were clustered to study the species composition diversity of the samples of different groups, and the OTUs were the sequences that were clustered with 97% consistency (Identity). Species annotations to the representative sequence of OTU with GreenGenes 13_8 database to obtain a relative abundance of the phylum level. It is shown here that at least a phylum that has a relative abundance in a sample is greater than 1%. The index Chao 1, Shannon and inverse Simpson were used to reflect the alpha diversity of the sample. Chao1 predicts the number of microorganisms based on the number of OTUs in the measured samples, which is also a standard for measuring species abundance. Shannon and Simpson are the diversity index that integrates OTU abundance and OTU uniformity.

## 3. Results

### 3.1. MIC Distribution and Establishment of the ECV

As shown in Appendix A, all *MG* isolates have been verified by Polymerase Chain Reaction (PCR). To ensure the accuracy and reliability of experimental procedure, *Escherichia coli* (*E. coli*) (ATCC 25922) was selected as the Quality Control (QC) strain. The MIC of danofloxacin for *E. coli* (ATCC 25922) was 0.03 µg/mL, which was within the acceptable QC range (0.008~0.06 μg/mL) according to CLSI document M37–A3. The MICs of 111 isolated *MG* strains ranged from 0.008 to 8 μg/mL, including 0.008 µg/mL (0.90%), 0.015 µg/mL (1.80%), 0.03 µg/mL (1.80%), 0.06 µg/mL (4.50%), 0.125 µg/mL (6.30%), 0.25 µg/mL (33.33%), 0.5 µg/mL (22.52%), 1 µg/mL (16.22%), 2 µg/mL (9.91%), 4 µg/mL (1.80%), 8 µg/mL (0.90%). The MIC_50_ and MIC_90_ were 1 µg/mL and 2 µg/mL, respectively.

Using the ECOFFinder software, the fitted MIC distribution of danofloxacin against *MG* was determined (Figure 1). The ECV at 95%, 97.5%, 99.0%, 99.5% and 99.9% confidence intervals were 1 μg/mL, 1 μg/mL, 2 μg/mL, 2 μg/mL and 4 μg/mL, respectively (Appendix A). The MIC of 95% confidence rate or 97.5% was selected as the ECV. Accordingly, the ECV of danofloxacin against *MG* was 1 μg/mL.

### 3.2. CO_PD_ for Danofloxacin against MG

#### 3.2.1. Pharmacodynamics of Danofloxacin against *MG* in the Plasma and Lung

M19 was chosen as the representative since it was a highly pathogenic strain as verified by the chicken embryo virulence test (Appendix A). The MICs of M19 to danofloxacin in FM-4 broth and lung tissue (healthy or infected) were 2 μg/mL. The MBCs in FM-4 broth lung tissue (infected) was 4 μg/mL, and 2 μg/mL, respectively. The data above indicated that the antimicrobial activity of danofloxacin in lung tissue was equivalent to that in FM-4 broth.

The killing-time curves of danofloxacin against M19 in vitro are shown in Figure 2, and the killing-time curves of danofloxacin against M19 in the healthy plasma, the healthy lung and the infected lung are shown in Figure 3, Appendix A, respectively. The bactericidal effect of danofloxacin against *MG* in vitro was similar to that of ex vivo. These data demonstrated that danofloxacin had an activity of killing that depends on the concentrations ex vivo, consistent with the in vitro bactericidal activity. The inhibitory efficiency gradually strengthened following the increased drug concentration when danofloxacin concentrations were higher than MIC (2 μg/mL). During 0–12 h, the danofloxacin concentration in the lung was higher than in the plasma. It revealed that danofloxacin had good distribution in the lung tissue and was suitable for pulmonary infections. At each time point, the danofloxacin concentration of the infected group was higher than that of the healthy group, which indicated that the disease had an impact on the distribution of the danofloxacin. The killing-time curves revealed that the antibacterial effect of danofloxacin against *MG* was concentration-dependent. The best PK–PD parameter for danofloxacin was the Area Under Curve/Minimum Inhibitory Concentration (AUC/MIC).

#### 3.2.2. Verification of HPLC Method for Determination of Danofloxacin

The linear range for the standard curve of danofloxacin ranged from 0.029 to 5 μg/mL (r^2^ = 0.9994) in plasma and 0.05 to 5 μg/mL (r^2^ = 0.999) in lung tissue. The limits of quantification (LOQ) were 0.05 μg/mL in plasma and 0.029 μg/mL in lung tissue. The limits of determination (LOD) were 0.027 μg/mL in plasma and 0.014 μg/mL in lung tissue. The mean recovery of danofloxacin was >87.62% in the plasma and lung tissue. The coefficient of variability (CV%) was <2.84% for both intra- and inter-day variation.

#### 3.2.3. PK Parameters of Danofloxacin in Plasma and Lung

The mean drug-time curve of danofloxacin (Figure 4A,B) in chicken lung tissues (healthy or diseased) includes the distribution phase and absorption phase, which accords with the first-order absorption two-compartment model. The pharmacokinetic data simulated by Winnonlin software are shown in Table 2.

The results shown that the concentration of drug in lung tissue was about 13–16 times higher than that in the plasma (Appendix A). As displayed in Table 1, the peak time (T_max_) was 0.99 ± 0.10 h, and the peak concentration (C_max_) was 0.28 ± 0.01 µg/mL, the AUC was 6.12 ± 0.24 h·µg/mL in infected plasma; in infected lung tissue, T_max_ was 1.30 ± 0.01 h, C_max_ was 4.71 ± 0.29 µg/mL, AUC was 23.67 ± 2.24 h·µg/mL. Since the T_max_ in plasma is lower than 1/2 MIC of the selected strain, it was considered that the bactericidal effect could not be achieved. Therefore, the PK parameters in the target diseased lung tissue were selected to determine the CO_PD_ of danofloxacin against *MG*.

#### 3.2.4. Integration Modeling

The parameter values of the E_max_, E_0_, N (Hill coefficient), and AUC_24h_/MIC of the model for three different levels of the growth inhibition conditions are shown in Table 3. The standards of the AUC_24h_/MIC ratio calculated for the E = 0 (bacteriostatic activity), E = −3 (bactericidal activity), and E = −4 (bacterial elimination) were 20.09, 37.34, and 46.67 h, respectively.

#### 3.2.5. Monte Carlo Simulation and CO_PD_

As is shown in Appendix A, the study analyzed the CO_PD_ corresponding to different E values. This study was mainly based on the PD target value when E = −3, which killed 99.9% of the bacteria to achieve clinical treatment to develop a drug resistance standard. Therefore, the CO_PD_ of danofloxacin against *MG* in the lung tissue of this subject was 0.5 μg/mL, at which the probability of target attainment (PTA) ≥ 90%.

### 3.3. CO_CL_ for Danofloxacin against MG

#### 3.3.1. Dose Regiment Based on the PK–PD Model

The protein binding rate of danofloxacin with different drug concentrations in lung tissues remained basically the same; both were 10%. The dose of (AUC_24h_/MIC)_ex_ corresponding to the purpose of different antibacterial effects of danofloxacin was substituted into the dose calculation equation to determine the dose required to achieve different antibacterial effects (Appendix A). In summary, the 24 h dosing interval of 16.60 mg/kg b.w. of oral administration for three consecutive days was the optimal dosing regimen, as determined by MLXPLORE software.

#### 3.3.2. Establishment of the Standard Curve of *MG* by Real-Time Fluorescence Quantitative PCR

The initial copy number of the plasmid constructed in this experiment was 1.58 × 10^9^ copies/μL. After PCR amplification of the standard plasmid, the amplified products were sequenced. The obtained sequences were compared using Blastn (NCBI) and showed that the homology between the *MG* S6 lipoprotein gene and the target fragment was 99.3%, indicating that the recombinant plasmid was successfully constructed. The amplification curves and standard curve of the quantitative fluorescence PCR in this experiment are shown in Appendix A, where the logarithm of the plasmid copy number is taken as the abscissa (X), and the Ct value was taken as the ordinate (Y). The formula was Y = −3.334X + 42.791, the correlation coefficient was R^2^ = 0.990, and the amplification efficiency value was 99.5%. The real-time fluorescent quantitative PCR method can detect the lowest concentration of plasmid standards at five copies/μL and no amplification curve for other bacteria (*Salmonella* ATCC 14028, *Klebsiella pneumoniae* CMCC 46117, *Pseudomonas aeruginosa* ATCC 27853, *Staphylococcus aureus* ATCC 6538, *Pasteurella pneumotropica* ATCC 35149, *Corynebacterium kutscheri* ATCC 15677), which indicated the good sensitivity and specificity. Use this fluorescence quantitative PCR method to detect a variety of common pathogens. The results are shown in Appendix A. Substituting the Ct value of the PCR assay into the standard curve equation, the initial copy number of clinical sample DNA could be calculated.

#### 3.3.3. Clinical Efficacy Assessment Indicators and Bacteriological Prognosis

The clinical treatment outcome is shown in Table 4. It could be seen from the results that as the MIC value of the infected strain increases, the clinical cure rate, degree of air sac damage, and weight gain were also significantly reduced.

DNA was extracted from the throat swabs of the chicken during all the experiments to calculate the bacterial load (Appendix A). The number of bacteria in the experimental group increased with an increase in infection time. When the administration started, the number of bacteria decreased significantly, and the therapeutic effects of low MIC challenge were better than that of chickens with high MIC, according to most of the observations. Only the group challenged with the high MIC *MG* strain could be detected on the third day, but the copy number was close to the detection limit.

#### 3.3.4. Determination of CO_CL_

The obtained data were first analyzed by the WindoW method, and the parameter values MaxDiff (the method of maximum difference) and CAR (the cumulative area under the curve (AUC) ratio) were calculated as shown in Table 5 and Appendix A, respectively. The AUC_Total_ for MIC = 4 μg/mL was calculated as [0.5 × (0.016 × 15)] + [0.5 × (0.25 − 0.016) × (15 + 15)] + [0.5 × (1 − 0.25) × (15 + 15)] + [0.5 × (2 − 1) × (15 + 15)] + [0.5 × (4 − 2) × (15 + 15)] = 44.763; the AUC succ for MIC = 4 μg/mL was calculated as [0.5 × (0.015 × 15)] + [0.5 × (0.25 − 0.016) × (15 + 14)] + [0.5 × (1 − 0.25) × (10 + 14)] + [0.5 × (2 − 1) × (9 + 10)] + [0.5 × (4 − 2) × (8 + 9)] = 1.36, it was also equal to A + B + C + D + E. By this approach, the CO_CL_ was determined to be from 0.25 to 2 μg/mL. The CART (the classification and regression tree) results suggested that the CO_CL_ should be less than 0.63 μg/mL (Appendix A). Then the CO_CL_ was analyzed with a logistic regression model with the MIC as the independent variable (X axis) and the Probability of Cure as the dependent variable (Y axis) (Appendix A). The nonlinear regression equation was Y = 71.708 − 9.463X + 0.453 X^2^ + 0.206 X^3^, and the correlation coefficient (R^2^) was 0.98. Non-linear regression analysis was performed that when the cure rate was 90%, the log_2_MIC was −1.9, so the CO_CL_ was calculated to be less than 0.27 μg/mL. The CO_CL_ were analyzed by the WindoW approach, nonlinear regression and CART analysis were 0.25 μg/mL–2 μg/mL, 0.25 μg/mL–0.63 μg/mL and 0.27 μg/mL, respectively. Combining the three CO_CL_ value ranges obtained, the 0.25 µg/mL was selected as eligible CO_CL_.

#### 3.3.5. Evaluation of the Effect of Danofloxacin on the Lung Microbiota

As is shown in Figure 5, 36 phyla were identified at the phylum level for the four groups in total, and 14 phyla were accounted for 1% with an abundance of more than 1%. The number of *Bacteroidetes*, *Firmicutes*, and *Proteobacteria* were dominant flora in four groups. Meanwhile, *Firmicutes* was the most prevalent phylum with an abundance between 39.08% and 79.77%. Moreover, *Bacteroidetes* and *Proteobacteria* were accounted for 12.41~21.46% and 4.07~31.90%, respectively. During the trial period, *Firmicutes* had a great decrease after challenge with *MG* and an obvious recovery after treatment with danofloxacin. A significant difference (increased) was observed for *Proteobacteria* for the post-treatment group. *Bacteroidetes* was no obvious change among the four groups. From the perspective of bacteriological classification, *MG* belongs to *Firmicutes*. The number of *Firmicutes* decreased with the *MG* infected group because there may be competition for the growth between *MG* and other *Firmicutes*. It decreased *MG* and recovery for the other *Firmicutes* with the danofloxacin treatment group. *Firmicutes* reached a similar level with the normal state after cure. *Bacteroides* have no noticeable change for the four groups since danofloxacin is more effective for treating Gram-positive bacteria, *Mycoplasma,* and some anaerobic bacteria. However, *Bacteroides* belongs to Gram-negative bacteria. As for the decrease of *Proteobacteria* after the challenge with *MG* and the recovery after treatment with danofloxacin, there was some evidence that danofloxacin is more effective for *E. coli* [43] and *Salmonella* [44], which belong to the *Proteobacteria*.

To identify the common and unique OTUs (Operational Taxonomic Units) among different samples, the Venn diagrams were generated to make qualitative comparisons of the common OTUs between groups and the unique OTUs for each group (Figure 6). Representative sequences for each OTU were retrieved and classified to the genus level. There were 3584, 2894, 2852, and 2382 OTUs detected in the HWK (control group), GW (infection group), ZW (treatment group), and ZHW (post-treatment group) group, respectively. From the Venn diagram (Figure 6), a significant change in the respiratory tract flora was seen in chickens after *MG* infection. The number of respiratory tract flora decreased significantly after the cure by danofloxacin. 

However, when we focused on the unique flora of each group, we found that the unique flora in the control group was the highest. The number of unique flora in the respiratory tract decreased gradually with the *MG* challenge and the use of danofloxacin and post-treatment. The study shows that the lung microbiota colonized in the respiratory tract decreased after the challenge of *MG* because the immunity was reduced. *MG*, as a dominant strain, occupies a large amount of space in the lungs and competes with other flora for territory food, resulting in a decrease in the number of unique flora. There was some potential target pathogen of danofloxacin in the respiratory tract after the infection of *MG*. It also proved the fact that danofloxacin was a broad-spectrum antibacterial drug. In addition, all kinds of bacteria groups were reached a minimum value because of the use of danofloxacin and the enhancement of chicken immunity.

The Chao 1, inverse Simpson, and Shannon are indices that represent the alpha diversity of microbial communities. The Chao 1 (richness) index shows the number of different taxa present in the sample. However, the indices inverse Simpson and Shannon (diversity) show the difference among the abundance of other taxa. As is shown in Appendix A, the control group had the highest average values of Chao 1, followed by the danofloxacin-treated group, the *MG* infected group, and the post-treatment group. The results show that the number of species of bacteria decreased significantly after the challenge with *MG*, and the number of bacterial species increased when treated with danofloxacin. However, the number of strains reduced to a minimum value when *MG* was cured. The average values of microbial diversity indices of the three processing groups were consistently lower than the control group. The studies show that administration of danofloxacin to chicken affected the bacterial taxa in the chicken respiratory tract and resulted in decreased richness and diversity of microbiota.

## 4. Discussion

The *MG* infection is transmitted both horizontally and vertically. It is said that the three main strategies for the control of *MG* infections include, biosecurity and management practices, vaccination, and treatment [45]. Acquisition of fertile eggs and chicks from *MG* free sources, regular surveillance and monitoring of the flocks culling of the carrier birds, and strict biosecurity measures are needed to prevent the infection in the poultry flocks [46]. For immunization, inactivated, live attenuated and recombinant vaccines are commercially available for *MG* [47]. Antibiotics that inhibit the metabolic processes of microorganisms, especially fluoroquinoles, are commonly used to treat the infection [48]. Reasonable treatment is recommended to control CRD caused by *MG* to minimize their economic losses. Although antimicrobials should be used prudently to slow the emergence of resistance, little research has been performed to establish the ECVs, CO_PD_, and CO_CL_ of antibacterial agents against *MG*. It has been proven that early danofloxacin treatment was very effective against *MG* [15,16].

Several methods have been reported for the establishment of ECVs. For example, Rodriguez-Tudela et al. [49] determined ECVs by twofold dilutions above the modal MIC. Arendrup et al. [50] performed ECVs by twofold dilution steps higher than the MIC_50_. Paterson and Turnidge [51] estimated ECVs by statistical analysis. The statistical method is a useful tool and has been widely used for the establishment of ECVs. The study used nonlinear regression analysis to determine the ECVs, and the results obtained were the same as those simulated by ECOFFinder software.

A suitable drug susceptibility method could be selected according to the growth characteristics of bacteria [52]. Previous research suggested that the results of drug sensitivity tests obtained by broth dilution methods based on the CLSI guidance document and literature are valid data [53,54]. Using the micro-dilution method, the MIC values for danofloxacin were 0.008–8 μg/mL for the *MG* isolates, which proved different dilution multiples [55]. In this study, the ECV of danofloxacin against *MG* was 1 μg/mL. A total of 111 *MG* isolates, representing a wide range of strains, were collected throughout China, which was considered to be a wide range of sources and representation. It was reported that the MIC range of *Mycoplasma capricolum* (*M. capricolum*) against danofloxacin was 0.1–6.4 μg/mL, and the susceptible breakpoint of danofloxacin against *M. capricolum* was 0.25 μg/mL [56].

The MIC range, MIC_50_, and MIC_90_ values for the antimicrobial susceptibility of *Mycoplasma hyopneumoniae* to enrofloxacin were 0.12–16 μg/mL, 0.25 μg/mL, 4 μg/mL, respectively [57]. It was also reported that the MIC range of *M**ycoplasma pneumoniae* isolates to levofloxacin was 0.125–1 μg/mL [58]. The results above revealed that our study is generally consistent with the results of previous studies. However, the ECV of danofloxacin against *MG* had not yet been established. It is said that as for a microbiological point of view, both empirical and targeted antimicrobial treatment in respiratory infection is based on the sensitivity profile of isolated microorganisms and the possible resistance mechanisms that they may present, which may vary in different geographic areas according to prescription profiles and vaccination programs [59]. Some *MG* isolates with higher MIC for danofloxacin were found in our study, which indicated temporal and geographic differences in the prevalence of resistance.

Pharmacokinetic data, MIC distribution, and PK–PD target were essential parts of the establishment of the CO_PD_. Researchers reported that tilmicosin, doxycycline, valnemulin, tiamulin and tylosin have the rationality of AUC/MIC as the PK–PD parameter [60,61,62,63,64]. Furthermore, the drug concentrations in the target sites were directly correlated with clinical efficacy, so the researchers chose the lung tissue in infected chicken as the object to establish CO_PD_ [65,66]. The selection of PK–PD parameters have been reported in the literature. Xiao et al. [9] found that danofloxacin showed a significant concentration-dependent bactericidal effect on *Salmonella* Typhimurium, so the AUC/MIC parameters were selected for the PK–PD model parameters. Yang et al. [67] studied the PK–PD of danofloxacin on *E. coli* in the piglet ileum ultrafiltrate and found that the correlation and simulation effect of the parameter AUC/MIC as a PK–PD parameter is better than the parameter T (Time) > MIC that represents the cumulative time that the concentration exceeds the MIC. Xu et al. [13] fitted the PK–PD model very well by selecting the AUC/MIC in vivo as the parameter. Bonassa et al. [68] successfully predicted the cure success rate of pathogens under different MICs through Monte Carlo simulation and PK–PD parameters (AUC/MIC). Fernández-Varón et al. [69] found that the PK–PD surrogate markers of efficacy AUC_24_/MIC and the Monte Carlo simulation show that marbofloxacin with subcutaneous is adequate to treat contagious agalactia affected by *Mycoplasma agalactia* (*M. agalactia*) in lactating goats. Zhang et al. [11] reported the PK–PD relationship of danofloxacin to *MG* in the in vivo infection model by the PK–PD parameter AUC/MIC. Accordingly, our study chose AUC/MIC as the PK–PD parameters for anti-*Mycoplasma* effects. 

The data in Appendix A revealed that the average danofloxacin concentration in lung tissues was higher than the corresponding concentration in plasma (13–15 times higher), which is consistent with previous results [70]. Because danofloxacin can be accumulated at the infection site (lung), the CO_PD_ in plasma may not represent the significant parameters of the lung. The pharmacokinetic studies of danofloxacin on chickens are relatively few. Zhang et al. [70] reported that the C_max_, T_max_, and AUC_24_ of danofloxacin in lung tissues at a dose of 5 mg/kg b.w. in *MG* infected chickens was 1.14 ± 0.15 μg/mL, 2.67 ± 1.15 h, and 10.26 ± 2.42 μg·h/mL, respectively. It is different from the C_max_ (4.71 ± 0.29 μg/mL), T_max_ (1.30 ± 0.01 h), and AUC_24_ (23.67 ± 2.24 μg·h/mL) in our study. The differences in PK parameters among different studies may be due to differences in chicken breeds, and individual differences in experimental animals are also important reasons. Danofloxacin easily penetrates the tissue, so the drug concentration of danofloxacin in lung tissue was higher than that in plasma [11]. The C_max_ could be reached in plasma and lung tissue for about 1 h following oral administration, indicating that danofloxacin absorbs rapidly in plasma and lung tissue. The infected groups had a higher drug concentration distribution at the target site, showing that danofloxacin is suitable for the treatment of deep tissue diseases [71].

The trial data in our study showed that the success rate for treatment of *MG* with MIC of 0.25 μg/mL was 93.3% and of 0.016 μg/mL was as high as 100%. This study used a previously reported method [72] to establish a real-time PCR method to detect changes in the copy number of *MG* in clinical trials, with the detection limit of 158 copies, which was 1000 times more sensitive than a regular PCR method. With the increase of the infection time, the bacteria increasingly colonize the chickens, which is consistent with the changes in the clinical symptoms of the chickens. The trial results showed that the effective rate and cure rate of the M2 and M24 strains were more than 90%, because the strains were sensitive, and the drug had a better therapeutic effect. The effective rate and cure rate of the M57 strain drug treatment group were low. The possible reason for this is that M57 may contain acquired resistance genes or other resistance mechanisms, and requires further study.

The formulation of the clinical cut-off value mainly includes two aspects: the MIC distribution of bacteria and the data related to clinical efficacy. Because there was no standard method to establish CO_CL_, our study established the CO_CL_ mainly based on the following three approaches, which included the “WindoW” approach recommended by CLSI [27] and the nonlinear regression by VetCAST. The latter includes the formula of POC = 1/(1 + e^−a + bf (MIC)^) to calculate the relation between the dependent variable of POC and the independent variable of MIC [73]. Compared with four other supervised classifiers (J48, the OneR decision rule, the naïve Bayes classifier, and simple logistic regression), the CART method was recommended by Toutain et al. [24], and Cuesta et al. [74] could obtain the best statistical results. Bhat et al. [75] reported that the CO_CL_ of cefepime against gram-negative organisms was 8 μg/mL by CART software analysis.

There is a difference among the three cutoff values. In this study, the ECV, the CO_PD_, and the CO_CL_ were 1 μg/mL, 0.5 μg/mL, and 0.25 μg/mL, respectively. The ECV value was greater than the CO_CL_. From the MIC distribution of danofloxacin against 111 *MG* strains, the peak value was equal to the CO_CL_, and 0.5 μg/mL (CO_PD_) and 0.25 μg/mL (CO_CL_) account for 22.5% (25/111) and 16.2% (18/111), respectively. Overall, the difference was not significant. The result of ECV > CO_PD_ > CO_CL_ might indicate that there were clinically resistant bacteria, and the drug might be clinically ineffective against some clinical bacteria. Further studies are necessary to confirm the relationship between MIC phenotype and resistance genotype.

The respiratory microbiota is a complex community and widely recognized in many aspects of research [76,77,78], but little is known on the change between the challenge with *MG* and treatment with danofloxacin. It is reported that the composition of the lung microbiota is increasingly well characterised, with changes in microbial diversity or abundance observed in association with chronic respiratory diseases [79]. 16S rRNA sequence data analysis showed that the chicken’s respiratory tract contained various bacterial species. It is consistent with the report that *Firmicutes*, *Proteobacteria* and *Bacteroidetes* were the prevalent flora in lung microbiota [36]. The results suggest that the number of lung microbiota decreased gradually in the trial period, but there was no significant difference between the *MG* infected group and the treatment group. The OTU Venn diagram shows that the control group has the highest unique flora, and with a huge decrease in the infection group, the unique flora reached the minimum value with the treatment of danofloxacin. This trend might be caused by the reaction of the bacterial stress response [80]. It indicated that danofloxacin was effective against *MG* and killed many other bacteria in the chicken respiratory tract.

A variety of physiological processes related to bacterial survivability and colonization ability could be decreased under the stress response, leading to the competitive exclusion by other commensal bacteria during the treatment period [81,82]. These genera may have significant potential targets for fluoroquinolone antimicrobial. Moreover, in the previous studies, the diminishing of taxa in response to antimicrobial treatment was also reported [83]. The results show that the respiratory tract’s richness and diversity were generally decreased during the infection and treatment, which is consistent with the previous study [35,77]. Consequently, the rational use of danofloxacin for the treatment of *MG* infections should consider the antibacterial effect and the influence of danofloxacin on the respiratory microbiota. Few pieces of the research reported the potential health impact of changes in the respiratory microbiota in chicken when treated with danofloxacin. However, there are adverse effects on the health due to disrupting the respiratory flora because of antibiotics.

The respiratory tract flora is in a state of dynamic equilibrium under normal circumstances; once the balance is broken, it is easy to cause an imbalance in the proportion of normal flora, leading to disease in animals, which is called “flora dysregulated”. Some dysregulation is only an abnormal change in quantity and generally has no clinical manifestations or only mild adverse reactions, which can be recovered naturally after drug withdrawal. However, antibiotics that were trialed by oral administration could affect autoimmunity by affecting gut microbiota colonization. Wang et al. [84] studied the correlation between gut microbiota and *MG* infection by disturbing gut microbiota in chickens with an antibiotic cocktail. The results showed that gut microbiota dysbiosis impairs pulmonary immune response against *MG* infection. The studies above have shown that antibiotics are a double-edged sword, so more attention should be paid to the rational use of drugs.

## 5. Conclusions

The present study firstly established the ECV (1 µg/mL) at 95% confidence intervals, CO_PD_ in lung (0.5 µg/mL) and CO_CL_ (0.25 µg/mL) of danofloxacin against *MG*. The ECV value was higher than CO_PD_ and CO_CL_, indicating that certain *MG* isolates may resist danofloxacin. The ECV, CO_PD_, and CO_CL_ values could be considered preliminary cutoff values. According to the CLSI M37-A3 decision tree (Appendix A), danofloxacin’s final clinical breakpoint (CBP) against *MG* is 1 µg/mL. These results provide a standard and reference for the use of danofloxacin in the treatment of *MG*. In addition, the therapeutic regimen obtained from PK–PD model is 16.60 mg/kg b.w. with a 24-h interval oral administered for three days. The research shows that the effect of drug treatment on lung microbiota should be considered when one therapeutic drug is needed.

## Figures and Tables

**Figure 1 antibiotics-11-00403-f001:**
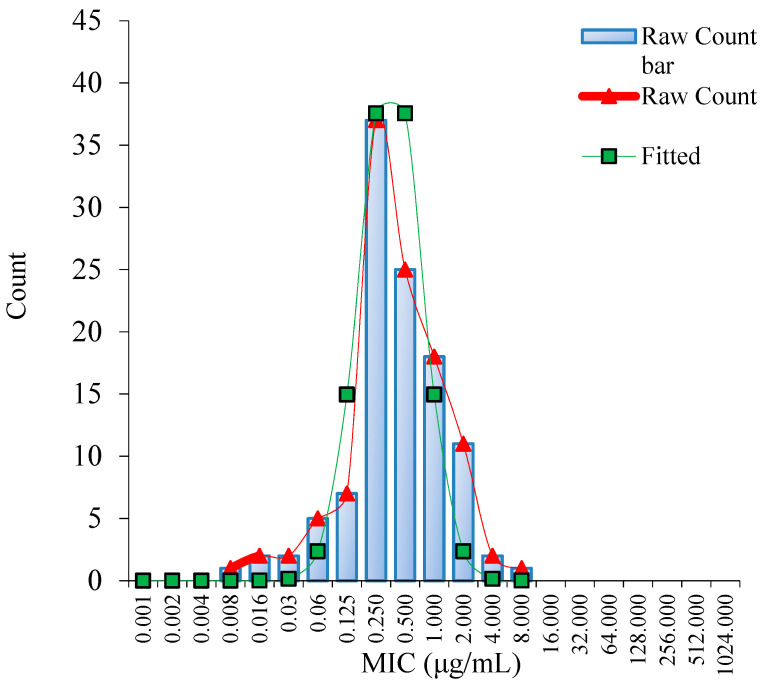
Nonlinear regression of MIC distribution of danofloxacin against *M**G* (*n* = 111). Note: “Raw Count” was the measured MICs of danofloxacin; “Fitted” was the simulated MICs of danofloxacin.

**Figure 2 antibiotics-11-00403-f002:**
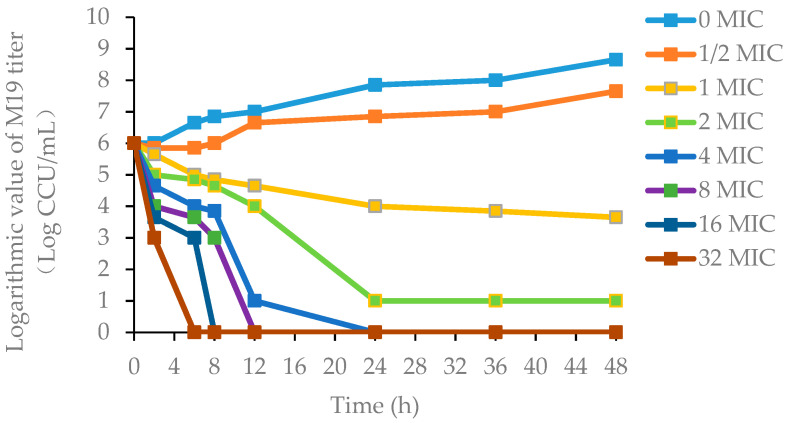
The killing-time curves of danofloxacin against M19 in FM-4 broth (in vitro). Note: M19 is a highly pathogenic strain as determined by the chicken embryo virulence test. FM-4 is a medium for the culture of *MG*. MIC (2 μg/mL) is the minimum inhibitory concentration of danofloxacin against M19.

**Figure 3 antibiotics-11-00403-f003:**
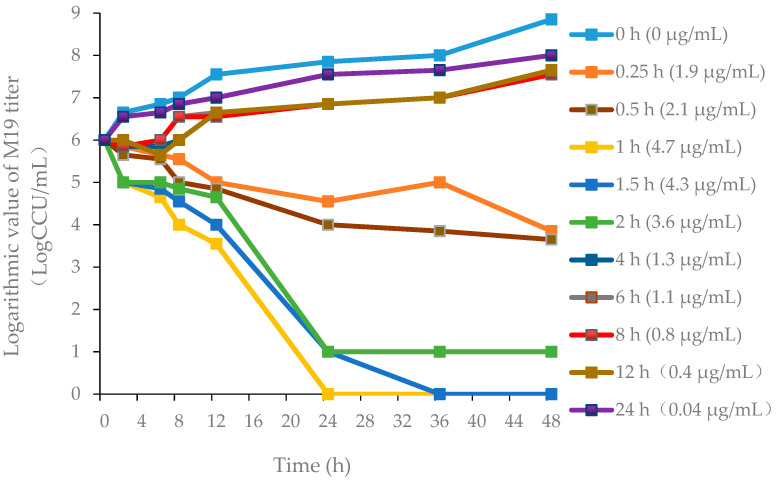
Ex vivo killing-time curves in infected lung tissue.

**Figure 4 antibiotics-11-00403-f004:**
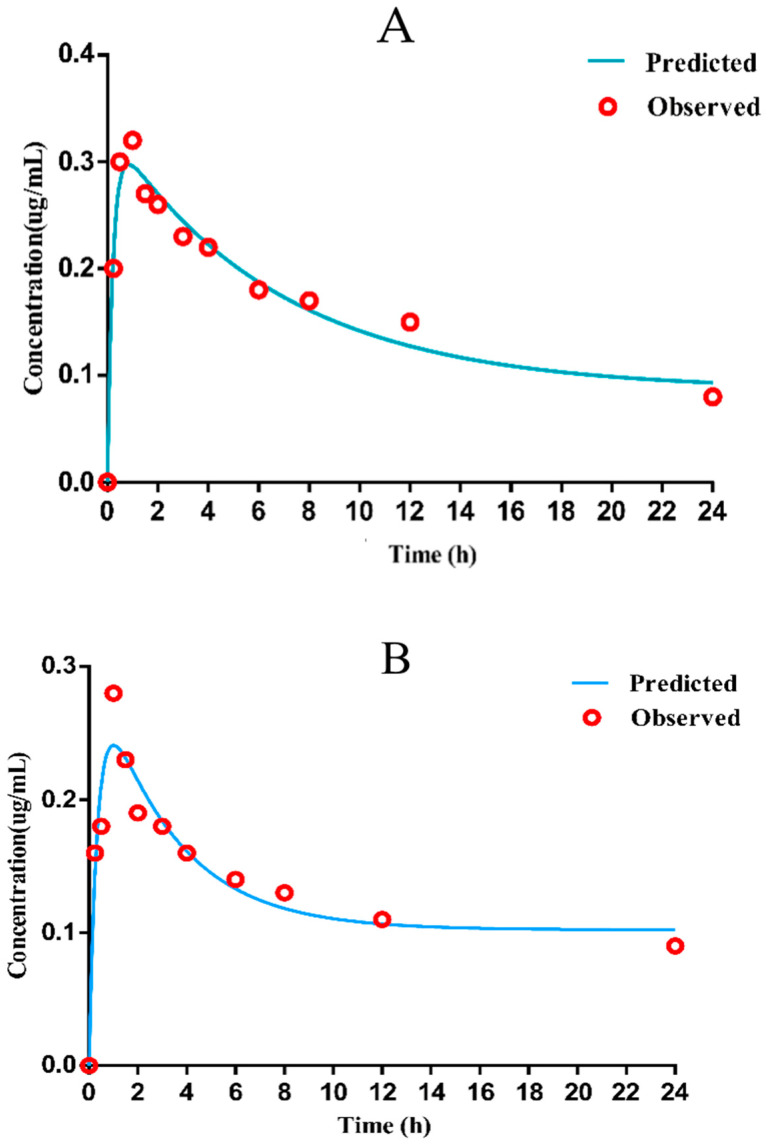
Mean concentration versus time for danofloxacin in healthy and diseased lung ((**A**): healthy lung (**B**): diseased lung).

**Figure 5 antibiotics-11-00403-f005:**
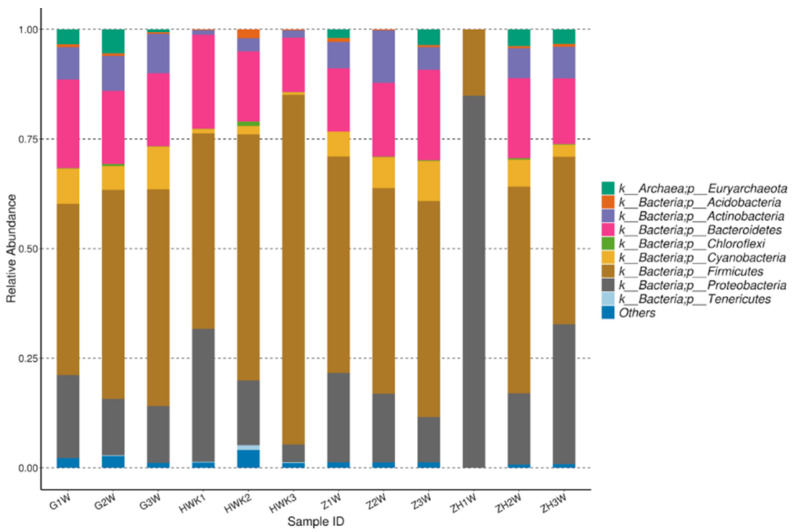
Taxonomic classification of the 16S rRNA sequences at phylum. Note: HWK represents the control group; GW represents the infection group; ZW represents the treatment group, and ZHW represents the post-treatment group.

**Figure 6 antibiotics-11-00403-f006:**
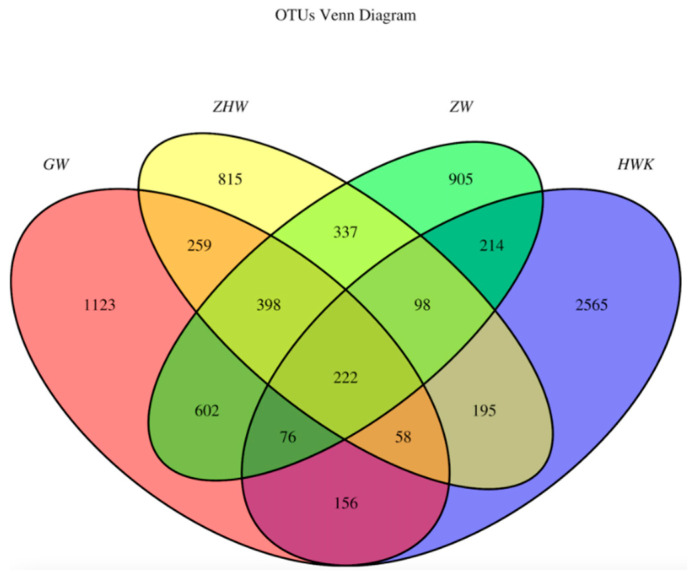
Venn diagrams of the common and unique OTUs of the four groups. Note: HWK represents the control group; GW represents the infection group; ZW represents the treatment group, and ZHW represents the post-treatment group. The numbers in the Venn diagram indicate the common (overlapping part) and unique OTUs of each sample group. The common OTUs represent the overlapping part between two or more groups, and the unique OTUs represent the part where no overlap occurred.

**Table 1 antibiotics-11-00403-t001:** Primer sequences for identification of *MG*.

Primer	Sequence	Amplicon Size
*MG*-13R	GAGCTAATCTGTAAAGTTGGTC	185 bp
*MG*-14F	GCTTCCTTGCGGTTAGCAAC	

**Table 2 antibiotics-11-00403-t002:** PK parameters in plasma (left) and lung (right) in healthy (*n* = 65) and infected (*n* = 65) broilers after an oral administration of danofloxacin at a dose of 5 mg/kg.

Parameter	Units	Plasma		Lung	
		Healthy Group	Infected Group	Healthy Group	Infected Group
C_max_	μg/mL	0.32 ± 0.00	0.28 ± 0.01	4.28 ± 0.17	4.71 ± 0.29
T_max_	h	0.82 ± 0.04	0.99 ± 0.10	1.28 ± 0.03	1.30 ± 0.01
α	1/h	1.80 ± 0.02	1.32 ± 0.03	0.75 ± 0.02	0.76 ± 0.02
β	1/h	0.05 ± 0.02	0.03 ± 0.01	0.05 ± 0.01	0.07 ± 0.01
T_1/2α_	h	0.38 ± 0.02	0.53 ± 0.13	0.92 ± 0.03	0.91 ± 0.03
T_1/2β_	h	14.55 ± 0.83	25.21 ± 0.72	13.77 ± 0.43	9.77 ± 0.72
CL/F	(mg/kg)/(h·μg/mL)	0.93 ± 0.08	0.82 ± 0.06	0.23 ± 0.02	0.22 ± 0.05
AUC_24_	h·μg/mL	5.40 ± 0.46	6.12 ± 0.24	21.97 ± 2.80	23.67 ± 2.24

Note: C_max_ is the peak concentration; T_max_ is the peak time; α is the distribution rate constant; β is the elimination rate constant; T_1/2α_ is the distribution half-life; T_1/2β_ is the elimination half-life; CL is the clearance rate, F is the bioavailability, CL/F is the body clearance corrected by bioavailability, and AUC is the area under the curve of the drug.

**Table 3 antibiotics-11-00403-t003:** The Sigmoid Emax model of danofloxacin in the lung.

Parameters	Units	Diseased Group
E_max_	LgCCU/mL	1.85
E_0_	LgCCU/mL	−6
EC_50_	h	31.22
N	−	2.67
E_max_ − E_0_	LgCCU/mL	7.85
(AUC_24h_/MIC)_ex_E = 0	h	20.09
(AUC_24h_/MIC)_ex_E = −3	h	37.34
(AUC_24h_/MIC)_ex_E = −4	h	46.67

Note: (AUC_24h_/MIC)_ex_E is the difference of antibacterial CCU logarithm of lung tissue samples incubated with danofloxacin; E_max_ is the maximum difference of antibacterial CCU logarithm of lung tissue samples incubated with danofloxacin; E_0_ is the difference after 24 h incubation in CCU antibacterial logarithm in control samples; EC_50_ is considered the PK–PD parameter value ex vivo when producing 50% maximal bactericidal effect in the lung tissue sample; N is the Hill coefficient, which is used to determine the S-shaped curve, describe the effect value E linearization ex vivo and the slope of the PK–PD parameter value.

**Table 4 antibiotics-11-00403-t004:** The effect of danofloxacin for the treatment of different MIC of *MG*.

Group	MIC (μg/mL)	Total	Valid	Cure	Effective Rate (%)	Cure Rate (%)	Average Gain	Air Sac Average Damage Score	Air Sac Injury Reduction Rate (%)
Blank group	-	15	-	-	-	-	290.00 ± 68.10	0.00 ± 0.00	100
M57 Test group Control group	4	15	13	8	86.7	53.3	185.10 ± 50.20	2.57 ± 0.50	58
		15	-	-	-	-	100.21 ± 28.20	2.90 ± 0.80	0
M19 Test group Control group	2	15	12	9	80	60	230.10 ± 25.20	1.73 ± 0.30	62
		15	-	-	-	-	97.10 ± 35.20	2.87 ± 0.77	0
M73 Test group Control group	1	15	11	10	73.3	66.7	255.50 ± 29.22	0.71 ± 0.70	64
		15	-	-	-	-	80.30 ± 19.30	3.00 ± 0.52	0
M24 Test group Control group	0.25	15	14	14	93.3	93.3	247.20 ± 32.35	0.56 ± 0.72	79
		15	-	-	-	-	59.10 ± 14.00	2.93 ± 0.69	0
M2 Test group Control group	0.016	15	14	15	93.3	100	268.10 ± 18.23	0.53 ± 0.73	85
		15	-	-	-	-	80.30 ± 16.39	3.2 ± 0.93	0

**Table 5 antibiotics-11-00403-t005:** The CAR and MaxDiff results of WindoW analysis for five strains of different MIC.

Strain	MIC (μg/mL)	Success Treatment	%Success ≤ MIC	%Success > MIC	MaxDiff	AUC_Succ_	AUC_Total_	CAR
M2	0.016	15	100.00 A	68.33 B	31.67	0.12	0.12	1
M24	0.25	14	93.30	60.00	33.30 C	3.51	3.63	0.967
M73	1	10	66.67	56.67	10.00	12.513	14.88	0.841
M19	2	9	60.00	53.33	6.67	22.013	29.88	0.737
M57	4	8	53.33	53.33	0.00	39.093	59.88	0.653

Note: A is Subset A, B is Subset B, and C is MaxDiff. Subset A = 100 × (15/15); Subset B = 100 × (14 + 10 + 9 + 8)/(15 + 15 + 15 + 15).

## Data Availability

The data presented in the study are deposited in the National Center for Biotechnology Information (NCBI) repository, the accession number is PRJNA771179.

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
