# Peer review of "Rational Use of Danofloxacin for Treatment of Mycoplasma gallisepticum in Chickens Based on the Clinical Breakpoint and Lung Microbiota Shift"

_antibiotics, 2022, doi:10.3390/antibiotics11030403_

Round 1
Reviewer 1 Report
I am delighted to review this manuscript, covering an important subject i.e treatment of Mycoplasma in Chicken with danofloxacin and influence on the lung microbiota. The manuscript follows the scope of the journal Antibiotics. A detailed study has been done with an excellent presentation of research work.
I would recommend the article could be published in Antibiotics with minor corrections. And the authors need to address the below-mentioned queries.
- Space is missing for “55°C” and for all temperatures.
- Change “ml” to “mL” wherever applicable.
- The author could divide figure 2 into four parts for better representation.
- In the supplementary. Note: Figure S4A and S4B should be S3A and S3B.
- The article is lengthy, the author could try to reduce it by avoiding common facts in the introduction and discussion.
- Some parts of the result could be moved to supplementary.
- The author could include the following relevant references.
- Arzey GG, Arzey KE. Successful treatment of mycoplasmosis in layer chickens with single dose therapy. Aust Vet J. 1992 Jun;69(6):126-8. doi: 10.1111/j.1751-0813.1992.tb07478.x. PMID: 1379426.
- Huang A, Wang S, Guo J, Gu Y, Li J, Huang L, Wang X, Tao Y, Liu Z, Yuan Z, Hao H. Prudent Use of Tylosin for Treatment of Mycoplasma gallisepticumBased on Its Clinical Breakpoint and Lung Microbiota Shift. Front Microbiol. 2021 Sep 9;12:712473. doi: 10.3389/fmicb.2021.712473. PMID: 34566919; PMCID: PMC8458857.
- Xu, Z.; Huang, A.; Luo, X.; Zhang, P.; Huang, L.; Wang, X.; Mi, K.; Fang, S.; Huang, X.; Li, J.; et al. Exploration of Clinical Breakpoint of Danofloxacin for Glaesserella parasuis in Plasma and in PELF. Antibiotics 2021, 10, 808. https://doi.org/ 10.3390/antibiotics10070808
Author Response
Reply:
Dear Editor,
Thank you very much for your comments. These comments are very helpful. I have carefully revised every part of the article according to the comments. Corresponding revisions are shown in red Mark in the text.
For the comment 1, the space between number and °C were checked and added for the whole article;
For the comment 2, We made some adjustment according to the advice that the figure 2 were divided into four parts (figure 2, figure3, figure S2 and figure S3), and we put the figure S2 and figure S3 in the supplementary material to avoid to be lengthy.
For the comment 3, we revised the number of the revelant figures.
For the comment 4, we made some revisions and deleted the comment facts in the introduction and discussion. In the discussion, we deleted the sentences “MG is a curicial ……..farms [41]”, “However, the clincal …..established.”and so on. Deleted content is highlighted in yellow with a cut-out line “-”.
For the comment 6, we moved figure S2, S3, S6, S8, S9 and Table S5 to supplementary. We also made changes to the corresponding serial number changes in the article.
For the comment 7, we have cited the above references to the corresponding content in the article, whose serial number is 13, 66, 68.
Reviewer 2 Report
This article covers very important topic to poultry industry. Generally I appreciate the concept and description of the study.
There are few minor issues that need to be improved.
Abstract: In my opinion abstract is to long (>350 words), the journal guideliness suggest about 200 words. Please make it shorter.
Citation no 11 is highlighted in blue. Please change it.
There is different font in the references, please standarize it.
Discussion: Can you please discuss in few sentences (other than antimicrobial therapy) ways to manage MG infection in chicken flocks.
Author Response
Reply: Thank you very much for your affirmation of the article, and also thank you for your suggestions on revisions. The following are some of the revisions we made for the article. Corresponding revisions are shown in red Mark in the text.
For the comment 1, we concisely generalized the abstract to make it shorter (about 200 words).
For the comment 2, the color of citation no 11 were revised to black.
For the comment 3, we standrized the font in the references and 2.6.3 to be consistent in whole article.
For the comment 4, we added the revelant contents about “ways to manage MG infection in chicken flocks” in line 593-600.